# Combination Strategies of Different Antimicrobials: An Efficient and Alternative Tool for Pathogen Inactivation

**DOI:** 10.3390/biomedicines10092219

**Published:** 2022-09-07

**Authors:** Nagaraj Basavegowda, Kwang-Hyun Baek

**Affiliations:** Department of Biotechnology, Yeungnam University, Gyeongsan 38451, Korea

**Keywords:** multidrug resistance, bacterial infections, synergistic effects, antimicrobial agents, pathogen inactivation

## Abstract

Despite the discovery and development of an array of antimicrobial agents, multidrug resistance poses a major threat to public health and progressively increases mortality. Recently, several studies have focused on developing promising solutions to overcome these problems. This has led to the development of effective alternative methods of controlling antibiotic-resistant pathogens. The use of antimicrobial agents in combination can produce synergistic effects if each drug invades a different target or signaling pathway with a different mechanism of action. Therefore, drug combinations can achieve a higher probability and selectivity of therapeutic responses than single drugs. In this systematic review, we discuss the combined effects of different antimicrobial agents, such as plant extracts, essential oils, and nanomaterials. Furthermore, we review their synergistic interactions and antimicrobial activities with the mechanism of action, toxicity, and future directions of different antimicrobial agents in combination. Upon combination at an optimum synergistic ratio, two or more drugs can have a significantly enhanced therapeutic effect at lower concentrations. Hence, using drug combinations could be a new, simple, and effective alternative to solve the problem of antibiotic resistance and reduce susceptibility.

## 1. Introduction

The rapid emergence and spread of multidrug-resistant (MDR) bacteria has become a serious global public health threat [1]. Long-term exposure and increased use and abuse of antibiotics could result in bacterial tolerance, which renders them less effective or even ineffective, and the mechanism includes changing the targets of antibiotics [2]. MDR species are not only restricted to hospitals or healthcare environments; they are also found in humans, animals, plants, food, water, soil, and air. Moreover, they can be passed from person to person and between animals and persons. Antibiotic resistance is observed in various extracellular, intracellular, pathogenic, and nonpathogenic bacterial species. Among Gram-positive MDR species, *Staphylococcus aureus*, *Streptococcus pneumoniae*, *Enterococcus faecium*, and *Enterococcus faecalis* are the most common. Among Gram-negative strains, *Pseudomonas aeruginosa*, *Escherichia coli*, *Klebsiella pneumoniae*, and *Acinetobacter baumannii* are the most common MDR species [3]. However, methicillin-resistant *Staphylococcus aureus* causes pneumonia, bacteremia, soft tissue infections, and other fatal diseases [4]. Similarly, multiple antibiotic-resistant *Acinetobacter* spp. and *Klebsiella* spp. are the most commonly reported.

Recent studies suggest that biofilm-associated infections account for more than 65% of all infections, and antibiotics lack effectiveness against biofilm-associated bacteria [5]. Biofilms can shield bacteria from host defenses, disinfectants, antibiotics, and many antimicrobial agents. This leads to a reduced bacterial growth rate, decreased metabolic activity, and promotion of tolerance to antibiotics [6]. Moreover, the excessive use of antibiotics is often not tolerated by the host organism, whereas lower doses are ineffective. In addition, conventional antibiotics support antibiotic resistance in viable bacteria [7]. Pathogens growing in biofilms exhibit both adaptive resistance to all antimicrobial agents and the host immune system by 10- to 1000-fold compared to their free-living, planktonic counterparts [8]. Hence, there is urgent need to search for alternative, novel, efficient antimicrobial agents and more targeted treatment strategies to overcome antibiotic resistance. An alternative strategy currently in practice or under trials includes using different antimicrobial agents in combination to produce synergistic effects [9]. Combination therapy is an attractive and optional treatment because it represents potential adjuvant targets of non-overlapping signaling pathways and decreases the risk of developing cross-resistance [10].

Many plants have been used as sources of natural products to maintain good health, especially antimicrobial compounds [11]. Plants have evolved many alternative strategies against pathogens, which involve various phytochemicals, secondary metabolites, and other chemical compounds [12]. Bioactive compounds derived from plants, such as alkaloids, phenols, flavonoids, tannins, peptides, and other medicinally important compounds, are responsible for their antimicrobial ability against MDR pathogens [13]. Combining two or more plant extracts or their phytochemical components produces mutual antimicrobial enhancement, an unlimited pool of compounds, and the expansion or strengthening of their effects when combined as a multidrug [14]. Combinations of different drugs elicit several advantages over their use as individual moieties, including enhancing the effectiveness of other antimicrobial agents, reduction in dosage, fewer side effects, better synergistic effect, attack of multiple target sites, reduced risk, and exhibition of potent and rapid antibacterial effects against MDR pathogens [15]. The pharmacological effects of these combinations could be initiated by multiple mechanisms of action of herbal–herbal interactions.

Similarly, combining plant extracts or active phytochemicals with antibiotics improves their efficacy against resistant bacterial pathogens [16]. Synergism due to this combination helps minimize the minimum inhibitory concentrations (MICs) of these agents and reduces the economic cost and sensory impact [17]. Another strategic approach to combat MDR bacteria involves using essential oils (EOs) combined with conventional antibiotics or plant-derived phytochemicals. EOs have been widely used for their unique flavors; fragrances; and antibacterial, antioxidant, antifungal, anti-inflammatory, and anticarcinogenic properties [18]. Combining two or more EOs or their components or interactions between EOs and their components with antibiotics is a promising alternative strategy to increase their additive and synergistic antimicrobial effects. EOs and antibiotics, in combination, produce stronger bacterial inhibition compared to when they are individually administered because they target different pathways to create multifaceted effects against powerful bacterial defenses, consequently needing a decreased dose of each component [19]. The synergism between EOs and antibiotics may be attributed partly to the EO-induced permeabilization of the cell membrane, resulting in the immediate transport of antibiotics into the interior of the cell [20].

Antimicrobial nanomaterials represent another strategic approach to fighting MDR bacteria in clinical practice. Metal and metal oxide-based nanoparticles (NPs) have been widely investigated over the last decade, owing to their favorable chemical, physical, magnetic, electrical, thermal, optical, and biological properties [21]. Consequently, nanomaterials have emerged as new tools to combat deadly bacterial infections due to their specific features, such as size, shape, morphology, stability, and surface charge [22]. A combination of EOs and nanomaterials might establish functional materials with modified surfaces, improved inhibitory effects, and the ability to bind target microorganisms to achieve maximum synergistic performance [23]. Thus, combining nanomaterials with either EOs or plant extracts may improve their interaction with the bacterial cell membrane, thereby inducing the disruption, damage, and killing of bacteria [22]. This review highlights the effects of different antimicrobial agents and the synergistic effects of combinations of plant extracts, EOs, and nanomaterials. Furthermore, we discuss their antimicrobial activities, mechanisms of action, and future perspectives on using different combinations of antimicrobial agents.

## 2. Antibacterial Activities of Plant-Derived Compounds

Although different kinds of synthetic antimicrobial agents have been introduced to the market in many countries, natural medicine from plants might effectively treat certain diseases such as diarrhea, cold, labor pain, and dental diseases. Globally, approximately 60,000 plant species are used for medicinal purposes, of which approximately 28,000 are well-documented, and only 3000 are estimated to be traded internationally [24]. As a result, the search for herbal medicines with relevant biological activity has gained additional value as they are associated with fewer side effects and are much cheaper and affordable [25]. Plants usually produce two types of metabolites, primary and secondary, that can be found in extracts of their flowers, roots, leaves, bulbs, seeds, and bark (Figure 1). Primary metabolites are crucial for plant growth and development, whereas secondary metabolites are involved in plant defense, physiology, and environmental communication [26]. Secondary metabolites include many specialized and active compounds derived from primary metabolites. These compounds show promising results in controlling the development of resistance against bacterial pathogens, including MDR bacteria, and combating other bacterial infections. Plant secondary metabolites are classified into three categories on the basis of their biosynthetic origins: terpenoids, phenolics, and alkaloids [27].

### 2.1. Terpenoids

Terpenes are an extensive and diverse group of naturally occurring, highly enriched compounds of secondary plant metabolites. On the basis of the number of their isoprene structures or units, they are classified as monoterpenes, diterpenes, triterpenes, tetraterpenes, or sesquiterpenes. Terpenes are also called isoprenoids, and their derivatives that contain additional elements, such as oxygen, are usually termed terpenoids. Monoterpenes are the smallest terpenes, comprising two isoprene units. Monoterpenes contain volatile compounds found in EOs extracted from different flowers, fruits, and leaves and are commonly used in fragrances and aromatherapy. The antimicrobial properties of these compounds have been studied for two decades, and several studies have reported that thymol, carvacrol, eugenol, and menthol exhibit significant activity against many pathogens [28]. Geraniol and thymol have shown the most activity against *Enterobacter* species and *S. aureus* and *E. coli*, respectively [29,30]. Diterpenes are naturally occurring chemical compounds that contain active groups such as vitamin A. Phytol is an acyclic diterpene alcohol that acts as an antitumor, cytotoxic, and anti-inflammatory agent. Diterpenes also inhibit the growth of *Staphylococcus aureus*, *Pseudomonas aeruginosa*, *Vibrio cholerae*, and *Candida* spp. [31]. Triterpenes contain six isoprene units derived from mevalonic acid and have been shown to inhibit the growth of *Mycobacterium tuberculosis*. The combination of rifampicin and oleanolic acid has shown synergistic antibacterial effects against some pathogens [32]. Tetraterpenes are also known as carotenoids because beta-carotene is a yellow pigment in carrots. Similarly, yellow, orange, and red organic pigments are produced by plants, and these substances have effective antifungal and antibacterial properties [33]. Sesquiterpenes are the most diverse group of terpenoids, consisting of three units of isoprene with a lower vapor pressure than monoterpenes because of their high molecular weight. Farnesol, a natural sesquiterpene, demonstrated antibacterial activity against *S. aureus* and *S. epidermidis* [34].

### 2.2. Phenolics

Phenols are the simplest bioactive phytochemicals. They are monomeric components of polyphenols and acids with a single substituted phenolic ring and are typically found in plant tissues such as melanin and lignin. The components catechol, orcinol, tarragon, pyrogallol, phloroglucinol, pyrocatechol, resorcinol, and thyme are effective against viruses, bacteria, and fungi [35]. Both catechol and pyrogallol are hydroxylated phenolic compounds that are toxic against microorganisms; catechol has two hydroxyl groups, while pyrogallol has three. The microbial toxicity of phenolic compounds depends mainly on the number and position of their hydroxyl groups, as hydroxylation increases toxicity [36]. The presence of two or more hydroxyl groups located at ortho, para, or meta positions to each other is the key factor for their antimicrobial activity. The presence of a hydroxyl group at the meta position of thymol makes it a more effective antibacterial agent than carvacrol, which has a similar structure, whose hydroxyl group is in the ortho position.

### 2.3. Alkaloids

Alkaloids are cyclic-nitrogen-containing organic compounds that have various chemical structures. More than 18,000 alkaloids have been discovered and studied phytochemically from different sources. Alkaloids are grouped into several classes, as natural, semi-synthetic, or synthetic, on the basis of their heterocyclic ring systems and biosynthetic precursors [37]. Alkaloids have various pharmacological activities, including antitumor, antihyperglycemic, anti-allergic, antidiabetic, antihyperlipidemic, and antibacterial. Piperine, berberine, quinolone, reserpine, sanguinarine, tomatidine, chanoclavine, conessine, and squalamine are the most important alkaloids with potent antibacterial activity. Piperine isolated from *Piper nigrum* and *Piper longum* inhibited the growth of mutant *S. aureus* when co-administered with ciprofloxacin [38]. A list of plants whose parts were reported to have antimicrobial activity against various pathogens, as well as their corresponding mechanisms of action, are summarized in Table 1.

## 3. Antimicrobial Efficacy of EOs

EOs are aromatic, lipophilic, and complex mixtures of volatile secondary metabolites that are mainly obtained from different parts of plants, such as leaves, herbs, flowers, buds, fruits, twigs, wood, bark, roots, and seeds [73]. EOs are extracted using hydrodistillation or steam distillation. EOs are lighter than water, with a strong flavor and odor reminiscent of their plant origin. The chemical composition of EOs is highly complex, with the main components being flavonoids, flavones, flavonols, phenols, polyphenols, tannins, alkaloids, quinones, coumarins, terpenoids, polypeptides, and lectins [22]. These compounds show potential pharmacological activities such as hepatoprotective, anti-inflammatory, antioxidant, anticancer, antiseptic, insecticidal, anti-parasitic, anti-allergic, antiviral, and antimicrobial properties [74]. Essential-oil-based products are in high demand in aromatherapy; as flavor-enhancers in food, beverages, cosmetics, perfumes, soaps, plastics, and resins; and in the pharmaceutical industries [75]. EOs have more than 50 components; however, only two or three of them are the major components present in high proportions. The other minor components are present in low amounts.

The amount of the different components of EOs varies with the parts and species of plants, as they are chemically derived from compounds and their derivatives [76]. The major constituents of EOs are terpenes and terpenoids, while other important compounds include aromatic and aliphatic constituents. The volatile components of EOs include a variety of chemicals such as alcohols (such as menthol, borneol, nerol, and linalool), acids (such as geranic acid and benzoic acid), aldehydes (such as citral), esters (such as linalyl acetate, citronellyl acetate, and menthyl), ketones (such as carvone, camphor, and pulegone), hydrocarbons (such as α-pinene, α-terpinene, myrcene, camphene, and p-cimene), ketones (such as camphor, pulegone, and carvone), phenols (such as carvacrol and thymol), lactones (such as bergapten), and peroxides (such as ascaridole), all of which play major roles in the composition of EOs (Figure 2) [77]. EOs show strong antibacterial activity against various pathogenic bacteria, including MDR pathogens, by penetrating the membrane of bacterial cells and disrupting their cellular structure. The antibacterial effectiveness of EOs differs across plant species and target bacteria depending on their cell wall structure (Gram-positive or Gram-negative). The association of some major constituents of EOs, such as eugenol, thymol, carvacrol, carvone, p-cymene, terpinene-4-ol, and cinnamic aldehyde, which easily penetrate and split in the lipid membrane, could disrupt the cell membrane; prevent cellular respiration; and lead to the loss of cell membrane integrity, removal of cellular contents, and finally cell death [78].

EOs of *Thymus serrulatus* and *Thymus schimperi* were shown to possess strong antibacterial activity against *Lactobacillus* and *S. mutans*, with higher contents of thymol and carvacrol compounds reported to be the cause of this inhibition [79]. Similarly, EOs have been isolated from various parts of *Eugenia caryophylata*, such as the buds, leaves, and stems, with the main components being eugenol, β-caryophyllene, and eugenyl acetate. These EOs are effective against *S. aureus*, *E. coli*, *B. subtilis*, and *S. typhimurium* [80]. Likewise, tea tree EO has been reported to cause changes in the membrane permeability and mycelial morphology of *Monilinia fructicola* [81]. Furthermore, a recent study of lavender EO against *A. hydrophila*, *A. caviae*, *A. dhakensis*, *C. freundii*, *P. mirabilis*, and *S. enterica* showed the presence of the major compounds linalool and linalyl acetate [82]. In another study, winter savory EO exhibited the strongest inhibitory effect against clinical oral isolates of *Candida* spp., with thymol as the major compound [83]. It has been reported that among the commercially available EOs, such as anise, cinnamon, clove, cumin, laurel, Mexican lime, and Mexican oregano, oregano EO has the highest antibacterial activity against *S. typhimurium* and *E. coli*, with thymol as its major compound [84]. Some of the most important and active EOs, their major constituents, mechanisms of action, and antimicrobial potential against pathogenic microbes are summarized in Table 2.

## 4. Antimicrobial Nanomaterials

With the emergence of bacterial resistance and biofilm-associated infections, clinical research is needed to develop novel, effective, long-term antibacterial and biofilm-preventative agents. Metals have been extensively studied among the most promising novel antimicrobial agents [111]. Recently, metal-based nanomaterials have become the most extensively and rapidly emerging materials in the field of medicine. Different types of metallic NPs have demonstrated strong antibacterial activity in many recent studies [112]. Generally, NPs have fascinating characteristics, such as a high surface area-to-volume ratio, size, shape, and surface activity, and exhibit superior electrical, catalytic, and optical properties. Due to their unique properties, NPs have a more well-developed surface than their microscale counterparts, affecting their antimicrobial efficiency and effectiveness [113]. Similar to antibiotics, metals can selectively inhibit metabolic pathways by interacting with bactericidal activity and ultimately kill MDR bacteria [112]; however, cells deviate from metal transport systems and metalloproteins [114]. Hence, NPs showed noticeable antimicrobial activity against both Gram-negative and Gram-positive pathogens such as *E. faecalis*, *B. subtilis*, S. *epidermidis*, multidrug-resistant *S. aureus*, and *E. coli* strains.

Metal NPs such as Ag, Au, Cu, Zn, Ti, Ga, Al, and Pt [115] and metal oxide NPs such as CuO, MgO, ZnO, TiO_2_, NiO_2_, SiO_2_, and Fe_3_O_4_ are known to display various antimicrobial properties, which have been known and applied for decades [116]. In addition, graphene oxide (GO) and carbon nanotubes (CNTs), such as single-walled carbon nanotubes (SWCNTs) and multi-walled carbon nanotubes (MWCNTs), are also excellent candidates due to their antimicrobial activities (Figure 3). Recently, metal–organic frameworks (MOF) and metal sulfide nanomaterials, such as FeS-, Ags-, ZnS-, and CdS-MOFs and Zn-, Cu-, and Mn-based MOFs, have also been proven to have antibacterial activities [117]. Multimetallic NPs, particularly NPs formed by at least two metals, such as bimetallic, trimetallic, and quadrametallic NPs, display rich optical, electronic, and magnetic properties. The properties of multimetallic NPs, including size, shape, surface area, and zeta potential, enhance their interaction with bacterial cell membranes. They could disrupt cell membranes, produce reactive oxygen species (ROS), damage the DNA, induce protein dysfunction, and may be potentiated by the host immune system [23].

Metallic biopolymer-based nanocomposite systems are well-known candidates as antimicrobial nanomaterials. In particular, cationic chitosan-based NPs bind to anionic cell membranes, resulting in alterations to the cell membrane, leakage of intracellular compounds, and eventually cell death [118]. Antimicrobial peptides (AMPs) have attracted great interest because of their high biocompatibility and low probability of inducing bacterial resistance. AMP-conjugated nanomaterials can hinder the growth of pathogens and kill bacteria on the basis of the inherent action of typical combination strategies [119].

AgNPs are considered the most common antimicrobial agents that can destroy a wide range of Gram-negative and Gram-positive bacteria. Ag ions combine with disulfide or sulfhydryl groups of enzymes, disrupt normal metabolic processes, and ultimately lead to cell death [120]. The bactericidal efficacy of Au NPs might have a greater chance of penetrating the bacterial cell wall by generating holes, leading to increased permeability and higher oxidative stress within the cytoplasm. Similarly, ZnO NPs displayed vigorous antimicrobial activity by releasing Zn^2+^ ions and generating ROS, owing to their electrostatic interaction and internalization. In contrast, smaller ZnO NPs increased the interaction and abrasiveness of the bacterial cell wall [121]. Cu and CuO NPs showed excellent antimicrobial activity against different strains of bacteria by releasing Cu^2+^ ions and stimulating ROS production [122]. Various studies have revealed the visible-light-induced antibacterial properties of Fe-, Cu-, Ni-, and Ag-doped TiO_2_ NPs against *E. coli* and *S. aureus* [123,124]; however, TiO_2_ NPs adversely affect human cells and tissues, so their use remains limited. SiO_2_ NPs, especially mesoporous NPs, have attracted considerable attention because their properties, such as size, matrix, and surface functions, which can be tuned to improve their interaction with and penetration of biofilm-producing bacteria [125].

Compared to monometallic NPs, multimetallic NPs, such as bi-, tri-, and quadrametallic NPs, have gained great importance and interest due to their unique physical, chemical, electrical, optical, and catalytic properties and applications in different fields [126]. Multimetallic NPs can be altered or tuned by controlling their structure, morphology, and chemical composition to achieve strong synergistic interactions and performance [127]. When at least two metals are formed as NPs, combinatorial approaches, such as structural changes, deduction of the lattice parameters, and total electronic charge shift improvements, are expected [128]. Recently, bimetallic Ag/Cu and Cu/Zn [129], and trimetallic Cu/Cr/Ni [130] and CuO/NiO/ZnO [131] NPs have exhibited remarkably improved antimicrobial performance compared to monometallic NPs. Recent studies on the antimicrobial activities of metal and metal oxides, including mono-, bi-, and trimetallic NPs, against various bacterial strains and their respective mechanisms of action are shown in Table 3.

Core–shell quantum dots (CSQDs) are a new type of fluorescent antibacterial nanomaterial with unique physical and chemical properties. Owing to their high electron transfer, CSQDs produce a large number of free electrons and holes that accumulate ROS inside the cell, inhibiting their respiration and replication [167]. CSQDs exert antimicrobial effects by destroying cell walls, binding with genetic material, and inhibiting energy production. The antimicrobial activities of some CSQDs and their mechanisms of action are listed in Table 4.

Additionally, the antibacterial properties of graphene involve both chemical and physical modes of action. The chemical action is associated with oxidative stress generated by charge transfer and ROS, while the physical action is induced by the direct contact of graphene with bacterial membranes [175]. Similarly, CNTs are more effective and cost-efficient, exhibiting strong antimicrobial properties owing to their remarkable structure. This mechanism is based on the interaction of CNTs with microorganisms and the disruption of their metabolic processes, cellular membranes, and morphology. Table 5 summarizes the various reported antimicrobial activities of graphene and CNTs.

Dendrimers are macromolecules with highly branched tree-like dendritic structures, narrow sizes, relatively large molecular masses, and well-defined globular structures [183]. Dendrimers peripherally cationic and highly water soluble due to numerous peripheral hydrophilic groups compatible with water [184]. Dendrimers can incorporate biologically active agents in the interior or periphery; therefore, they serve as carriers of biologically active agents [185]. Antimicrobial polymers or their composites can prevent or suppress the growth of microbes on their surfaces or in the environment. Positively charged polymer surface groups are attracted to negatively charged cell membranes, leading to cell membrane damage and cell death [186]. A few recent publications on the antimicrobial activities of dendrimers and polymer nanocomposites are summarized in Table 6.

## 5. Synergistic Antimicrobial Activity of Plant Extracts, EOs, and Nanomaterials

A synergistic effect is a process in which chemical substances or biological structures interact or combine to create an effect greater than the sum of the effects of the individual components. Synergy is the concept wherein the performance of two or more antimicrobial agents is combined and the effects of such mixtures are greater than those of the separate or individual components, enhancing solubility. In recent years, the synergistic combination of different antimicrobials and plant extracts has been considered a unique strategy to increase the spectrum of antimicrobial activity of these substances and prevent the development of resistant strains [194]. Plant extracts, EOs, and nanomaterials are commonly used as antimicrobial agents for the treatment of many infectious diseases. However, these antimicrobials are not very effective against acute infections because they lack a standardized and clinically applicable pharmaceutical form. Consequently, various antibiotics have been discovered as synthetic antimicrobials; however, these drugs are highly toxic and have poor tolerability, so bacteria develop resistance against them. Hence, a possible approach to improve and enhance antibacterial activity is to use combinations of different antimicrobials. These combinatorial approaches can be used alone or with other antimicrobials against a wide range of pathogens [195]. Compared to the individual substances or components, multicomponent antimicrobials display increased antimicrobial activity; therefore, other molecules present in the antimicrobial agents could control the function of the main components and improve their synergistic effects. Moreover, combining different antimicrobials offers many advantages, including a reduction in dosage, fewer side effects, decreased toxicity, extensive antibacterial action, and the ability to attack multiple target sites with increased efficacy [196]. Some combinations of antimicrobial agents, such as plant extracts, EOs, antibiotics, and NPs, are summarized in Table 7.

Combinations of antimicrobial agents, such as EOs/EOs, plant extract/plant extract, and NPs/NPs, already have confirmed antimicrobial activities [203]. Ncube et al. evaluated the bulb and leaf extracts of three medicinal plants, independently and in combination, against *S. aureus*. Their results showed the strongest synergistic effect compared with the effects observed with individual extracts [12]. Similarly, a combination of *Bulbine frutescens* and *Vernonia lasiopus* plant extracts showed improved antimicrobial activity against *E. coli* [204]. Obuekwe et al. found the largest zones of inhibition against *S. aureus* using a combination of *Ocimum gratissimum* and *Ficus exasperate*, and *Bryophyllum pinnatum* and *Ocimum gratissimum* against *E. coli* [205]. Recently, EO–EO associations showed a synergistic effect against vancomycin-resistant enterococci (VRE), methicillin-resistant *S. aureus* (MRSA), and extended-spectrum β-lactamase (ESBL)-producing *Escherichia coli* [195]. A mixture of *R. abyssinicus* and *D. penninervium* EOs showed strong synergistic effects against MRSA and *P. aeruginosa* [206].

The synergistic antibacterial activities of cumin, cardamom, and dill weed EOs against *C. coli* and *C. jejuni* have been reported [91]. In a previous study, a combination of cinnamon and clove EOs showed synergistic antibacterial activity against foodborne *S. aureus*, *L. monocytogenes*, *S. typhimurium*, and *P. aeruginosa* [207]. Garza-Cervantes et al. examined the synergistic antimicrobial activities of silver in combination with other transition metals (Zn, Co, Cd, Ni, and Cu). Their results exhibited synergism since the antimicrobial effects of the combinations against *E. coli* and *B. subtilis* increased up to eightfold when compared to the individual metals [208]. Similarly, β-lactam is the most common bactericidal agent recommended for the treatment of several infectious diseases. However, the increasing emergence of β-lactam resistance due to β-lactamase enzyme production is one of the most serious public health threats. Hence, current clinical trials suggest the use of proper combinations of β-lactam and β-lactamase inhibitors [209]. β-Lactam inhibitors are associated with β-lactam antibiotics because they are hydrolyzed by β-lactamases, and their main objective is to protect the associated antibiotics. β-Lactam inhibitors prevent the hydrolytic action of β-lactam antibiotics by binding to the active site of β-lactamase enzymes [210].

## 6. Currently Available Conventional Antibiotics

Currently, there are 32 antibacterial agents in clinical development phases 1–3 targeting WHO priority pathogens, 12 of which have activity against Gram-negative pathogens. Since 2018, several new products have entered phase 1 trials, and two new recombinant topoisomerase inhibitors, zoliflodacin and gepotidacin, have moved from phase 2 to phase 3 trials. Similarly, lefamulin and relebactum moved from phase 3 trials to FDA approval, and omadacycline and eravacycline have moved from NDA submission to gaining FDA approval. An additional product of β-lactam (cefideocol) is more stable against a variety of β-lactamases and has activity against all three critical priority pathogens [211]. β-Lactams are well-established and widely used antibiotics, including penicillins, cephalosporins, carbapenems, and monobactams. β-Lactams interrupt cell wall formation and subsequently disrupt peptidoglycan biosynthesis. However, the emergence of bacteria that produce β-lactamase enzymes that hydrolyze β-lactam antibiotics has rendered many antimicrobial agents ineffective. β-Lactamases are a diverse class of enzymes produced by bacteria that break the β-lactam ring open, inactivating the antibiotic. Table 8 summarizes several mechanisms of resistance to different target drugs with different modes of action.

## 7. Antibacterial Mechanisms of Plant Extracts, EOs, and Nanomaterials

The antimicrobial mechanism of plant extracts is more strongly correlated with the levels of their constituent phenolic compounds, especially flavonoids and their derivatives. The interaction of polyphenols with lipid bilayers can trigger and disrupt plasma membrane function, change its permeability, and form small pores. These could lead to the leakage of cell components, altering the surface electrical charge potential and bacterial polarity, modifying membrane fluidity, delocalizing membrane lipids and proteins, as well as other phenomena responsible for antibacterial activity. These alterations can cause severe damage to the bacteria by partitioning their membrane and cell wall, interrupting DNA and RNA synthesis and function, disrupting normal cell communication, and preventing biofilm formation [217]. Some studies have shown that secondary metabolites of plant extracts, such as alkaloids, terpenoids, and phenolic compounds, interfere with enzymes and proteins of the microbial cell membrane and inhibit enzymes necessary for amino acid biosynthesis. Other studies have ascribed the inhibitory effect of these plant extracts to their hydrophobicity since they can react with proteins and mitochondria, disturbing their structures and altering their permeability [218].

The antibacterial mechanism of EOs does not comprise a single action; however, various biochemical and structural mechanisms are simultaneously engaged at multiple sites in the bacterial cell membrane and cytoplasm. The primary antimicrobial effects of EOs are correlated with an increase in membrane permeability and plasma membrane disruption. The bioactive components found in EOs, such as thymol, eugenol, and carvacrol, might attach to the cell surface and penetrate the target region, especially the phospholipid bilayer of the cell membrane [76]. It has been shown that EO accumulation can disrupt membrane integrity and membrane proteins, increase membrane permeability, induce the leakage of cellular contents, and reduce the intracellular ATP pool. This consequently leads to cytoplasmic coagulation and the denaturation of enzymes, inhibiting the synthesis of DNA and proteins required for bacterial growth. Furthermore, the sustained loss of metabolites and ions due to EO administration can further disturb bacterial metabolic processes, leading to cell death [219].

The bactericidal mechanism of nanomaterials mainly depends on the type of NPs used, such as metals, metal oxides, and other nanocomposites. NPs bind to the bacterial cell wall, form membrane-penetrating pores, and release metal ions due to deposition. The adhesion of nanomaterials and microbial cells can be achieved through electrostatic attractions, hydrophobic interactions, Van der Waals forces, and receptor–ligand interactions, leading to cell wall destruction [220]. Furthermore, the positively charged surfaces of nanomaterials could promote the attachment of negatively charged bacterial surfaces, which may exert and strengthen their bactericidal effect. In addition, the generation of free radicals and ROS can destroy the cell membrane, disrupting the antioxidant defense system and causing mechanical damage to the cell membrane. Thereafter, nanomaterials interact with important cellular organelles such as DNA, enzymes, ribosomes, and lysosomes, resulting in oxidative stress, changes in membrane permeability, heterogeneous alterations, electrolyte balance disorders, changes in gene expression, and protein deactivation [220]. Possible modes of action when combining plant extracts, EOs, and nanomaterials are illustrated in Figure 4.

## 8. Concluding Remarks and Prospects for Future Research

Although the pharmaceutical industry has introduced many new antibiotics, the increasing prevalence of serious clinical complications related to MDR pathogens is a great challenge for researchers, clinicians, and the pharmacological industries. Therefore, searching for the most promising novel antibacterial agents with alternative strategies to combat bacterial infections is an ideal solution to treat infections that threaten human health. The ultimate goal is to offer appropriate and effective antimicrobial drugs to infected patients. The use of combination therapies is an effective way to improve the treatment of many health conditions, prevent the development of MDR pathogens, and reduce the treatment duration. Combination therapies can target and interact in multiple pathways and exhibit greater therapeutic efficacy than single antimicrobial-agent-based therapies. Moreover, they have recently been regarded as promising, cost-effective, and potentially able to mitigate side effects to the body with lower drug concentrations. There is plenty of evidence to support the effectiveness of medicinal plants in the treatment of infectious diseases. However, very few studies have reported the synergistic effects of plant extracts and phytochemical combinations of herbal remedies.

Combinations of different plant species or mixtures of different phytochemicals have been shown to exert potential antimicrobial activity against several human pathogens with diverse mechanisms of action. The curative effects of plant extract combinations showed both intrinsic and antibiotic-resistance-modifying activities. Some plant extracts are not effective as antibiotic agents alone; however, when combined with other antibacterial plant extracts, their bioavailability and antibacterial activity are enhanced. Similarly, there is much evidence suggesting the antimicrobial effects of individual EOs against different pathogens in vitro, but very few studies have reported the effects of EO combinations. Compared to a single EO or its chemical constituents, combining more than two EOs can increase and improve their antimicrobial activities due to an increased diversity of components and multiple sites of action. Furthermore, many studies have reported the potential antimicrobial activities of different nanomaterials against MDR pathogens. However, only a small percentage of studies have discussed the synergistic effects of multimetallic NPs, such as bi-, tri-, and quadrametallic and metal oxide nanocomposites. The synergistic effects of these multimetallic NPs have attracted considerable attention, owing to their diverse and tunable physicochemical properties and favorable catalytic properties compared with monometallic NPs.

The present review describes the synergistic effects of plant extracts/plant extracts, EOs/EOs, and nanomaterials/nanomaterials as efficient alternative strategies for pathogen inactivation or infectious diseases. However, further research is needed to assess the synergistic effects of combinations of plant extracts/EOs, plant extracts/nanomaterials, and nanomaterials/EOs to achieve better antimicrobial results. In addition, more effort is required to investigate the synergistic effects of combinations of plant extracts/EOs/nanomaterials. Consequently, β-lactam/β-lactamase inhibitor combinations are more effective on the different bacterial species. Combinations comprising three different antimicrobial agents might have enhanced synergistic antimicrobial activity compared to the effects of a combination of only two antimicrobial agents. Furthermore, the concentrations of the plant extracts, EOs, nanomaterial types, proper dosage, and choice of materials are essential to maximizing their therapeutic benefit. It is also important to consider environmental issues, health and safety concerns, risk assessments, potential toxicity, and hazards before considering them novel antimicrobial agents. These new, modern, and creative therapeutic strategies may exert a critical synergistic effect and serve as alternatives to conventional antibiotics for controlling the spread of pathogens. Finally, we believe this review provides necessary information about the combination of different antimicrobial agents to produce synergistic effects for the control and treatment of a wide range of pathogenic infections and may also play an essential role in many medical applications.

## Figures and Tables

**Figure 1 biomedicines-10-02219-f001:**
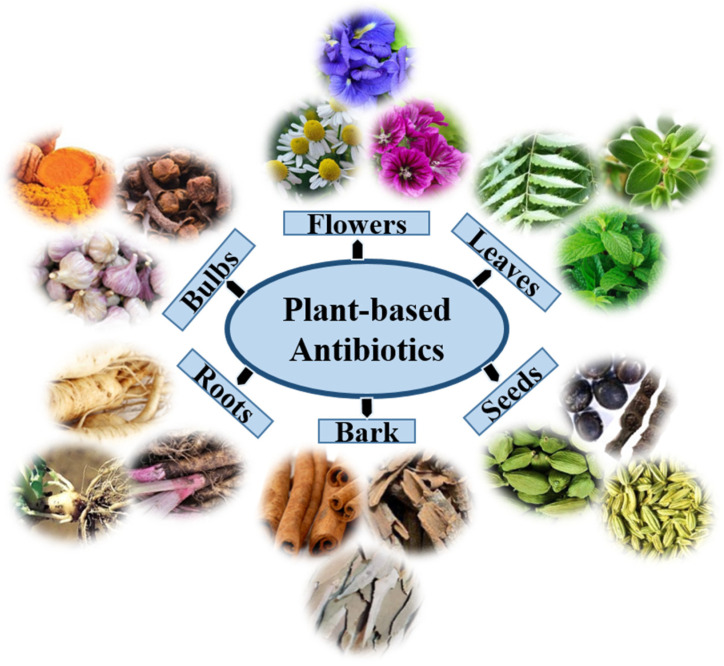
The extracts of plant organs, namely, the root, bark, bulbs, leaf, flower, and seed, may encompass distinctive phytochemicals with antimicrobial properties.

**Figure 2 biomedicines-10-02219-f002:**
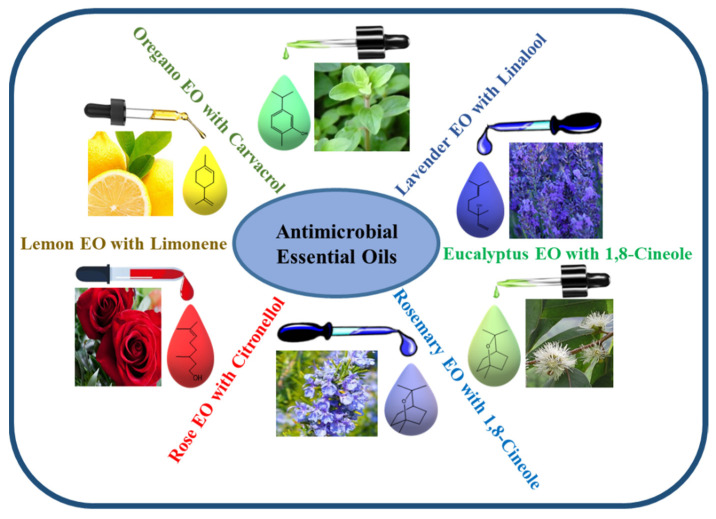
Schematic representation of some important EOs containing various components that have been screened for their antimicrobial properties.

**Figure 3 biomedicines-10-02219-f003:**
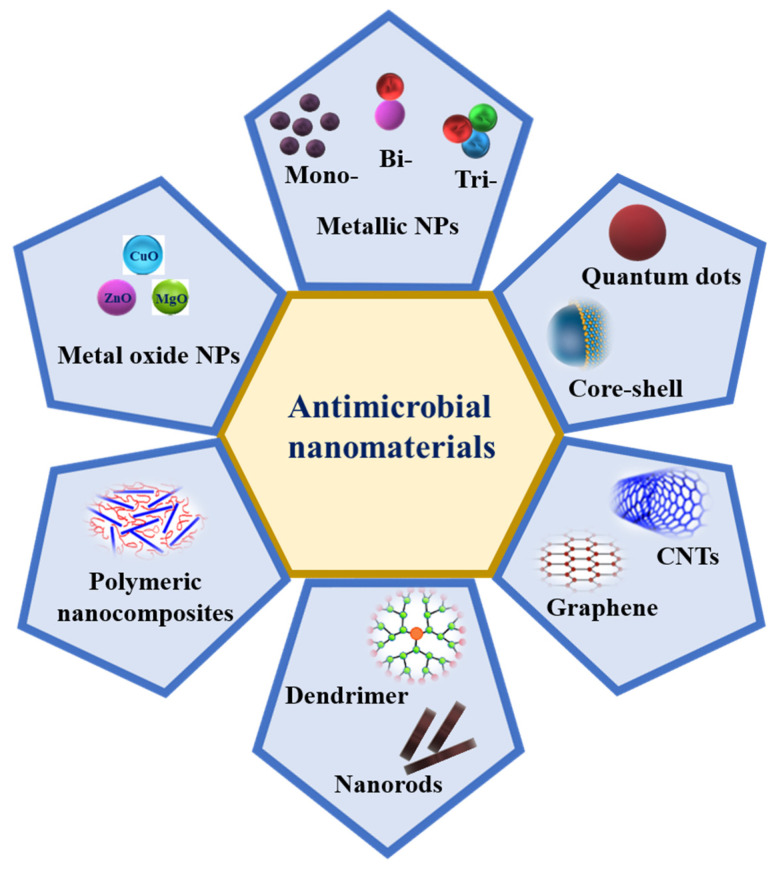
Schematic representation of different nanomaterials that possess antimicrobial activity.

**Figure 4 biomedicines-10-02219-f004:**
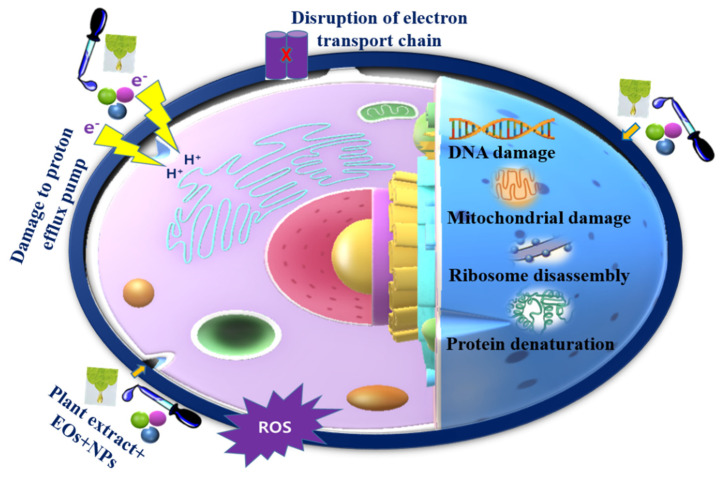
Proposed antibacterial mechanisms of combinations of plant extracts, EOs, and nanomaterials.

**Table 1 biomedicines-10-02219-t001:** A list of plants whose parts have been reported to have antimicrobial activity against various pathogens, as well as their corresponding mechanisms of action.

Plants	Parts	Pathogens	Mechanism	Ref.
*Alchornea cordifolia*	flower	*E. coli*	damage of cell wall	[39]
*Origanum majorana*	leaves	*S. aureus*, *K. pneumoniae*	membrane damage	[40]
*Psidium guajava*	leaves	*B. subtilis*, *S. aureus*	cell wall damage	[41]
*Justicia flava*	leaves	*E. coli*, *P. aeruginosa*	changes in internal pH	[42]
*Allium sativum*	bulbs	*P. aeruginosa*, *S. aureus*	cell membrane integrity	[43]
*Lannea welwitschii*	leaves	*E. coli*, *P. aeruginosa*	cell wall integrity	[42]
*Eucalyptus camaldulensis*	leaves, bark	*S. aureus*, *B. subtilis*	leakage of cell constituents	[44]
*Matricaria chamomilla*	flowers	*S. aureus*, *P. aeruginosa*	cell wall degradation	[45]
*Mentha piperita*	leaves	*S. aureus*, *B. subtilis*	damage of cytoplasmic membranes	[46]
*Foeniculum vulgare*	seeds	*A. flavus*, *C. albicans*	cellular DNA damages	[47]
*Melissa officinalis*	leaves	*S. aureus*, *P. aeruginosa*	disrupt the membrane structure	[48]
*Arctium lappa*	roots	*P. aeruginosa*, *S. aureus*	damage by oxidative stress	[49]
*Malva sylvestris*	flower, leaves	*S. aureus*, *E. faecalis*	damaging the membrane	[50]
*Thymus vulgaris*	leaves	*E. coli*, *S. aureus*	chemical affinity for membrane lipids	[51]
*Syzygium aromaticum*	buds	*E. coli*	membrane damage and intracellular content leakage	[52]
*Tribulus terrestris*	leaves	*Escherichia coli*, *Salmonella*	membrane damage and leakage of cellular materials	[53]
*Cinnamomum zeylanicum*	bark	*S. aureus*, *E. coli*	inhibiting of various cellular enzymes	[54]
*Zingiber officinale*	rhizome	*E. coli*, *S. aureus*	damage to cell membrane	[55]
*Curcuma longa*	rhizome	*S. aureus*, *B. subtilis*	loss of membrane integrity	[56]
*Eryngium foetidum*	leaves	*P. aeruginosa*, *C. albicans*	disruption of the cell membrane	[57]
*Portulaca oleracea*	roots	*E. cloacae*, *B. subtilis*	inhibiting the efflux pumps	[58]
*Momordica charantia*	peels	*S. aureus*, *B. cereus*	disintegrates the membrane	[59]
*Lawsonia inermis*	leaves	*S. aureus*, *E. coli*	inactivating microbial adhesions	[60]
*Azadirachta indica*	leaves	*S. pyogenes*	inactivating microbial enzymes	[60]
*Achyranthes aspera*	leaves	*S. pyogenes*	inhibiting energy metabolism	[60]
*Acacia nilotica*	seeds	*S. aureus*	cell membrane permeability	[61]
*Platanus hybrida*	fruits	*E. faecalis*, *E. faecium*	Inhibiting the biofilm production	[62]
*Cistus salviifolius*	aerial parts	*S. aureus*	cell wall alterations	[63]
*Punica granatum*	peels	*S. aureus*	cell wall alterations	[63]
*Piper betle*	leaves	*S. aureus*	destruction of the bacteria cell wall	[64]
*Ficus sycomorus*	leaves, fruits	*E. coli*, *S. aureus*	permeability of the cell membranes	[65]
*Myrtus communis*	leaves	*E. coli*	proteins in the outer membrane specifically involved	[66]
*Asphaltum punjabianum*	mineral resin	*E. coli*	proteins involved specifically in the outer membrane	[66]
*Marrubium vulgare*	leaves	*A. actinomycetemcomitans*, *E. corrodens*	affect cytoplasmic membrane	[67]
*Ocimum* *basilicum*	leaves	*P. aeruginosa*	bacterial cells will lose cations and macromolecules	[68]
*Clitoria ternatea*	flowers	*Streptococcus mutans*	quorum sensing inhibition	[69]
*Elettaria cardamomum*	Seeds	*P. gingivalis*	cell membrane disrupted	[70]
*Cinchona officinalis*	bark	*E. coli*, *P. aeruginosa*	structural damage of bacterial cells	[71]
*Panax ginseng*	roots	*B. cereus*, *S. aureus*	changes in the membrane potential	[72]

**Table 2 biomedicines-10-02219-t002:** Antimicrobial properties of various EOs with their respective plant sources and mechanisms of action.

Essential Oils	Plant Source	Major Components	Pathogens	Modes of Action	Ref.
Basil	*Ocimum basilicum*	linalool	*S. aureus*	disrupt the permeability barrier	[85]
Thyme	*Thymus vulgaris*	thymol	*P. aeruginosa*, *A. niger*	interferes with membrane functions	[86]
Clove	*Syzygium aromaticum*	eugenol	*S. aureus*, *S. Typhimurium*	sensitivity to eugenol	[87]
Cinnamon	*Cinnamomum zeylanicum*	cinnamaldehyde	*E. coli*, *L. innocua*	facilitate intracellular compounds leakage	[88]
Tea tree	*Melaleuca alternifolia*	terpinen-4-ol	*P. aeruginosa*, *C. glabrata*	alterations of the biological membrane	[89]
Rosemary	*Rosmarinus officinalis*	α-pinene	*C. albicans*	rupture of the membranes and cell wall	[90]
Dill	*Anethum graveolens*	carvone	*S. aureus*, *E. coli*	lesion in the plasma membrane	[91]
Cumin	*Cuminum cyminum*	p-mentha-1,3-dien-7-al	*S. aureus*, *E. coli*	deformation of the cell membrane	[91]
Cardamom	*Elettaria cardamomum*	α-terpinly acetate	*E. coli*, *S. aureus*	damage the cell membrane	[91]
Peppermint	*Mentha piperita*	menthol	*E. coli*, *S. aureus*	lysis and loss of membrane integrity	[92]
Anise	*Pimpinella anisum*	anethole	*S. aureus*, *B. subtilis*	alter the cell membrane permeability	[93]
Black pepper	*Piper nigrum*	α-pinene	*E. coli*	leakage, disorder, and death by breaking cell membrane	[94]
Sage	*Salvia officinalis*	α-thujone	*P. aeruginosa*	changed the cell membrane permeability	[95]
Lavender	*Lavandula angustifolia*	linalool	*S. aureus*, *E. coli*, *C. albicans*	damaging the cell wall and membrane	[92]
Mustard	*Brassica nigra*	allyl isothiocyanate	*A. fumigatus*, *A. nomius*	disrupt the cell wall thus causing cell lysis	[96]
Citron	*Citrus medica*	limonene	*S. aureus*, *E. coli*	destruction of the cell membrane	[97]
Eucalyptus	*Eucalyptus globulus*	1,8-cineole	*E. coli*, *S. aureus*	penetrate the membrane and damage cell organelles	[98]
Fennel	*Foeniculum vulgare*	trans-anethole	*S. aureus*, *E. coli*	cell deformation and integrity of cell membranes	[99]
Rose geranium	*Pelargonium roseum*	citronellol	*S. salivarius*	interaction with nitrogen in proteins and nucleic acids	[100]
Caraway	*Carum carvi*	carvone	*E. coli*, *B. bronchiseptica*	alteration in the structure of cell wall	[101]
Coriander	*Coriandrum sativum*	linalool	*S. tyhimurium*, *E. coli*	cell wall damage by over expression of genes	[101]
Turmeric	*Curcuma longa*	α-turmerone	*S. aureus*	inducing leakage of ions and important cell contents	[102]
Palmarosa	*Cymbopogon martinii*	geraniol	*B. subtillis*	alteration in cytoplasm and swelling	[103]
Dill	*Anethum graveolens*	α-phellandrene	*S. aureus*	disrupt the permeability barrier	[104]
Armoise	*Artemisia herba-alba*	thujone	*S. aureus*, *S. Typhimurium*	changing the membrane potential	[105]
Laurel	*Laurus nobilis*	1,8-cineole	*S. aureus. P. aeruginosa*	disrupt cellular membranes and increase membrane permeability	[106]
Ginger	*Zingiber officinale*	zingiberene	*S. aureus*, *E. coli*	destroy membrane structure, increase cell membrane permeability	[55]
Costmary	*Tanacetum balsamita*	β-thujone	*L. monocytogenes*, *S. sonnei*	damage to the cellular membranes	[107]
Guava	*Psidium cattleianum Sabine*	α-pinene	*S. aureus*, *N. gonorrhoeae*	propagate through cell membranes and cause the death	[108]
Marjoram	*Origanum majorana*	terpinen-4-ol	*S. aureus*, *K. clocae*	exhibited membrane and DNA damaging effects	[109]
Oregano	*Origanum vulgare*	thymol	*S. aureus*, *S. enterica*	alteration of the bacterial plasma membrane	[110]

**Table 3 biomedicines-10-02219-t003:** Antimicrobial activities of different metal and metal oxide nanomaterials against various pathogens and their respective mechanisms of action.

NPs	Size (nm)	Bacteria	Modes of Action	Ref.
Ag	10	*V. natriegens*	rupture of cell membrane and DNA damage	[132]
Ag_2_O	10	*L. acidophilus*, *S. mutans*	prevents the growth of pathogen	[133]
Ag_2_S	65	*Phormidium* spp.	cell membrane inhibition	[134]
Ag-MOF	-	*S. aureus*	stable in water and the existence of Ag^+^ ions	[135]
Al_2_O_3_	30	*S. typhi*, *F. oxysporum*	disintegration of outer membrane by ROS	[136]
Au	20	*S. pneumoniae*	cellular disruption	[137]
Bi	40	*M. arginini*, *E. coli*	inhibits protein synthesis	[138]
Cu	15	*B. subtilis*, *S. aureus*	synergistic effects of functional groups	[139]
CaO	58	*S. aureus*, *E. coli*	destruction of the cell membrane	[140]
CuO	60	*B. cereus*	damage of several biochemical processes	[141]
CeO_2_	5	*B. cereus*, *E. coli*	oxidative stress induced by the pro-oxidants	[142]
CdS	25	*S. aureus*, *Lactobacillus* sp.	CdS NPs impregnated and surrounded by the bacterial cell	[115]
Fe	474	*E. coli*	strong affinity between positively charged NPs and negatively charged cell membrane	[143]
Fe_3_O_4_	25	*E. coli*, *S. aureus*	plasma membrane disruption	[144]
FeS	35	*E. coli*, *S. aureus*	internalization of nanomaterials on cell membrane	[145]
Ga	305	*M. tuberculosis*	reduction of mycobacterium growth rate	[146]
Mn	50	*E. coli*, *S. aureus*	protein inactivation and membrane permeability decreases	[147]
MgO	27	*E. coli*, *Bacillus* sp.	loss of membrane integrity and leakage of intracellular molecules	[148]
Mn_3_O_4_	130	*P. aeruginosa*, *K. pneumonia*	disrupting bacterial cell membrane	[149]
Mg-MOF	-	*E. coli*, *S. aureus*	peptide–nalidixic acid conjugation formed	[150]
Mn-MOF	-	*E. faecalis*, *P. aeruginosa*	peptide–nalidixic acid conjugation formed	[150]
Ni	60	*P. aeruginosa*	destruction of cell membrane	[151]
NiO	40	*E. coli*, *B. subtilis*	oxidative stress generated at the NPs interface resulted in membrane damage	[152]
Pd	13	*S. pyrogens*, *B. subtilis*	cell membrane damage and apoptosis	[153]
Pt	2	*A. hydrophila*, *E. coli*	generation of ROS and decrease cell viability	[154]
Se	85	*S. aureus*, *E. coli*	ROS causing cell membrane damage	[155]
Si	90	*P. aeruginosa*, *S. aureus*	direct mechanical damage to the cell membrane	[156]
TiO_2_	9.2	*E. coli*	outer cell membrane damaged by attacking hydroxyl radicals and ROS	[157]
ZnO	30	*A. baumannii*	production of ROS increases	[158]
ZrO_2_	2.5	*S. mitis*, *S. mutans*, *R. dentocariosa*	NPs enhance the interaction with bacterial constituents	[159]
Zn-MOF	-	*P. aeruginosa*	causing cell damage by interaction with hydroxyl group of peptidoglycan	[160]
Ag/ZnO	43	*P. aeruginosa*, *S. aureus*	leaching of silver as Ag^+^	[161]
Au/CuS	2	*B. anthracis*	cell membrane damage	[162]
CuO/ZnO	50 and 82	*S. aureus*, *E. coli*	membrane depolarization caused due to lectrostatic interaction of NPs	[163]
Fe_3_O_4_/ZnO	200	*E. coli*, *S. aureus*	plasm membrane disruption includes oxidative stress	[164]
Au/Pt/Ag	20	*E. faecalis*, *E. coli*	ROS production	[165]
Cu/Zn/Fe	42	*E. coli*, *E. faecalis*	cell disruption by released ions	[166]

**Table 4 biomedicines-10-02219-t004:** Antimicrobial activities of core–shell quantum dots against various pathogens with their respective mechanisms of action.

NPs	Size (nm)	Bacteria	Modes of Action	Ref.
ZnS and CdSe/ZnS quantum dot	1.9	*E. coli*, *B. subtilis*	toxic composition of CdSe QDs demonstrating antimicrobial behavior	[168]
CdSe/CdS/ZnS multi-core–shell quantum dots	12–38	*K. pneumoniae*, *P. aeruginosa*	rupturing of the membrane wall and cause of the decay of bacteria	[169]
Ag-PdS/ZnS/CdS core–shell quantum dots	8	*S. saprophyticus*, *E. coli*	establishment of the catalyst–microorganism complex and a catalyst-related ROS	[170]
ZnSe@ZnS core–shell quantum dots	3.6 and 4.8	*E. coli*, *S. aureus*	high affinity towards the thiol groups of bacterial cell surface proteins	[171]
Peptide-loaded CdSe quantum dot	9 and 14	*E. coli*, *S. aureus*	AP loaded on CdSe NPs had a higher water solubility and bioavailability	[172]
P-doped carbon quantum dots	2.75–4.25	*E. coli*, *S. aureus*	cell walls wrinkled and broken	[173]
Ag@Ag_2_O core–shell	19–60	*P. aeruginosa*, *S. aureus*	blockage of DNA replication and repair processes	[174]

**Table 5 biomedicines-10-02219-t005:** Antimicrobial activities of graphene and CNTs against various pathogens and their corresponding mechanisms of action.

NPs	Size (nm)	Bacteria	Modes of Action	Ref.
rGO-TiO_2_	32	*E. coli*, *S. aureus*	improve the contact between TiO2 surface and bacteria	[176]
GO-ZnO	14–26	*E. coli*	induces ROS to kill the bacteria	[177]
GO-Cu_2_O	30	*E. coli*, *S. aureus*	copper ions react with cytoplasmic constituents	[178]
DMS-GO-DMA	--	*E. coli*, *S. aureus*	GO induces membrane stress on contact by disrupting and damaging cell membranes	[179]
MWCNT-LVX	--	*S. aureus*, *P. aeruginosa*	inhibition of bacterial DNA replication	[180]
F-MWNTs	--	*E. coli*, *S. aureus*	smaller diameter of MWNTs can endorse damage to cell membrane through the cell–surface interaction	[181]
Ag-doped ZnO on SWCNTs	12–15	*E. coli*, *S. aureus*	production of ROS on the interaction samples with bacterial membrane	[182]
Au-doped ZnO on MWCNTs	12–18	*E. coli*, *S. aureus*	the toxicity of carbon nanotube is mainly affected by diameter, length, and surface functional group	[182]

**Table 6 biomedicines-10-02219-t006:** Antimicrobial activities of dendrimers and polymer composites against various pathogens with their mechanisms of action.

NPs	Size (nm)	Bacteria	Modes of Action	Ref.
Van-PAMAM-AgNP dendrimers	--	*S. aureus*	heterofunctionalized Van-PAMAM-AgNP dendrimers for intra-cellular entry through the cell wall and bacterial killing	[185]
G4-PAMAM dendrimer	10	*E. coli*, *B. subtilis*	disrupting of the cell membrane function and inhibiting cell wall synthesis, nucleic acid synthesis, and protein synthesis	[187]
PAMAM-G7 dendrimer	20	*P. mirabilis*, *S. aureus*	dendrimers are mediated by disrupting the bacterial outer and inner membrane by terminal amine groups	[188]
Amino-acid-modified polycationic dendrimers	--	*P. aeruginosa*	loss of membrane potential, inhibition of biosynthetic pathways, and free radical production	[189]
Triclosan-loaded polymeric composite	--	*S. aureus*, *K. pneumoniae*	at high concentrations, triclosan destroys the bacterial membrane, leading to its death	[190]
PBAT/Cu-NPs	100–200	*A. baumannii*, *E. faecalis*	polymer and metal nanocomposites increase the number of ions released from the nanoparticles into the polymer matrix	[191]
Piperazine polymer nanocomposite	559.7	*E. coli*, *S. aureus*	nanoparticles are distributed within the suitable polymer matrix	[192]
PVA/GO/Ag nanocomposites	--	*E. coli*, *S. aureus*	physical interactions of the bacterial cell with the nanoparticle	[193]

**Table 7 biomedicines-10-02219-t007:** Antimicrobial activity of combinations of plant extract, EOs, antibiotics, and NPs against different pathogens.

Antimicrobial Agents	Combinations	Pathogens	Ref.
EOs/EOs	*Melaleuca alternifolia*/*Cupressus sempervirens*	*E. coli*	[195]
EOs/antibiotics	*Eucalyptus globulus*/oxacillin	*S. aureus*	[195]
EOs/NPs	*Lemongrass*/chitosan NP	*E. coli*, *S. aureus*	[197]
Plant extract/antibiotics	*Salvadora persica*/amoxicillin	*P. gingivalis*, *T. forsythia*	[198]
Plant extract/EOs	*Origanum vulgare*/carvacrol	*S. aureus*	[199]
Plant extract/NPs	*Vatica diospyroides*/Ag NPs	*S. aureus*, *B. subtilis*	[200]
NPs/antibiotics	AgNPs/fluconazole	*S. aureus*, *E. coli*	[201]
β-Lactam/β-lactamase inhibitor	amoxicillin/potassium clavulanate	*S. aureus*	[202]

**Table 8 biomedicines-10-02219-t008:** Mode of action of different classes of antibiotics with their resistance profiles and target bacteria.

Antibiotics	Specific Drug	Modes of Action	Resistance Profiles	Target Bacteria	Ref.
β-Lactams	Penicillin G, amoxicillin, cephalosporin C	Cell wall synthesis inhibition	Hydrolysis, efflux, altered target, reduced permeability	*S. aureus*, *P. aeruginosa*	[212]
Aminoglycosides	Streptomycin, gentamicin	Inhibition of translation and cell membrane synthesis	Modifying enzyme inactivation by phosphorylation	*P. aeruginosa*, *V. cholerae*	[213]
Tetracyclines	Minocycline, doxycycline	30S ribosomal subunit	Monooxygenation, ribosomal modification	Staphylococci, Streptococci	[214]
Glycopeptides	Vancomycin, teicoplanin	Peptidoglycan biosynthesis	Altered target	*S. haemolyticus*, *E. faecium*	[214]
Macrolides	Erythromycin, azithromycin	Inhibition of protein synthesis	Glycosylation, efflux, methylation	Streptococci, Staphylococci	[215]
Phenicols	Chloramphenicol	Inhibition of protein synthesis	Acetylation by chloramphenicol acetyltransferase	*B. subtilis*, *S. pneumoniae*	[216]
Rifamycin	Rifampin	Inhibition of nucleic acid synthesis	ADP-ribosylation, efflux	*V. cholerae*, *E. coli*	[213]
Quinolone	Ciprofloxacin, levofloxacin	Inhibitors of DNA synthesis	Altered DNA gyrase	*S. aureus*, *P. aeruginosa*	[212]
Cationic peptides	Polymyxin B, colistin	Disrupt membranes	Altered target, efflux	*E. coli*, *S. typhimurium*	[213]

## Data Availability

Not applicable.

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
