# Peer review of "Combination Strategies of Different Antimicrobials: An Efficient and Alternative Tool for Pathogen Inactivation"

_biomedicines, 2022, doi:10.3390/biomedicines10092219_

Round 1
Reviewer 1 Report
I think this review paper is well written, with adequate organized chapters and fruitful information about novel methods used for diminishing of the bacterial resistance . I think this article can be accepted for publication in its present form.
I have read the above-mentioned paper again, and I really have no additional suggestions for its improvement. The Authors gave a nice survey of proven/suggested mechanisms of antimicrobial action of essential oils, phytochemicals and nanoparticles. They have listed the known mechanisms of antimicrobial actions and cited the authors which have discovered it. In the last paragraph, the Authors have proposed the future research in which they suggest the combination of essential oils, phytochemicals and nanoparticles. They have listed possible problems and obstacles which can be expected. The paper is a good base for further investigations in this field. The references are, generally speaking, new, and cited papers published in the last five years. This is a review paper so it is hard to say it is novel, but it is significant because the Authors generated the latest knowledge about EO, phytochemicals and nanoparticles and their potential in MDR bacteria treatment.
Author Response
Dear Editor and Reviewers
Manuscript ID: biomedicines-1870450
Dear Editor and reviewers
Thank you very much for your kind consideration and invaluable efforts to review our manuscript. We are truly grateful for the meaningful comments and insights of the reviewers and editor. We have carefully revised all issues raised by the reviewers. We feel that our manuscript has been significantly improved from the reviewer’s creative and insightful suggestions. We would like to submit our fully revised manuscript after revised of all comments. All corrections made are highlighted in green color in the revised manuscript. Please see the response to reviewers’ comments on the succeeding pages.
Thank you very much for your consideration.
Sincerely Yours,
Kwang-Hyun Baek
Reviewer 1
Comment 1
I think this review paper is well written, with adequate organized chapters and fruitful information about novel methods used for diminishing of the bacterial resistance. I think this article can be accepted for publication in its present form.
I have read the above-mentioned paper again, and I really have no additional suggestions for its improvement. The Authors gave a nice survey of proven/suggested mechanisms of antimicrobial action of essential oils, phytochemicals, and nanoparticles. They have listed the known mechanisms of antimicrobial actions and cited the authors which have discovered it. In the last paragraph, the Authors have proposed future research in which they suggest the combination of essential oils, phytochemicals, and nanoparticles. They have listed possible problems and obstacles which can be expected. The paper is a good base for further investigations in this field. The references are, generally speaking, new, and cited papers published in the last five years. This is a review paper so it is hard to say it is novel, but it is significant because the Authors generated the latest knowledge about EO, phytochemicals and nanoparticles and their potential in MDR bacteria treatment.
Response
We are thankful for the reviewer's note. As suggested by the reviewer, we have checked the minor spell check and English revisions.

Reviewer 2 Report
On request of Biomedicines, I have revised the manuscript review titled “Combination Strategies of Different Antimicrobials: An Efficient and Alternative Tool for Pathogen Inactivation”, by Nagaraj Basavegowda and Kwang‐Hyun Baek.
The main scope of this work was to emphasize a therapeutic strategy which could be a new, simplest, alternative, and most effective way to solve the global issue of antibiotic resistance and reduce susceptibility, i.e. the combination therapy. To this end, the authors have first reviewed the various type of unconventional antimicrobial agents currently known, such as plants phytochemicals, essential oils (EOs) and their most promising constituents, and antimicrobial nanoparticles (NPs), the latter including metal NPs (Ag, Au, Cu, Zn, Ti, Ga, Al, Pt etc.), metal oxide NPs (CuO, MgO, ZnO, TiO2, NiO2, SiO2 and Fe3O4), graphene oxide (GO), carbon nanotubes (CNTc) (SWCNTs and MWCNTs), cationic dendrimer and polymers, metal-organic framework (MOF), metal sulfide nanomaterials, multimetallic NPs, cationic chitosan-based NPs, and antimicrobial peptides (AMP). Secondly, they have reported some case studies of combination and the main mechanisms of action of phytochemical, EOs and NPs.
GENERAL COMMENTS
Considering the importance of finding new and effective therapeutic strategies to counteract the currently untreatable infections sustained by MDR pathogens and by biofilms producing bacteria and the so-called biomaterials-associated nosocomial infections, the topic of the present review is original and interesting. Anyway, the way the authors have afforded the subject, is poor and badly organized. Essential parts, which in my opinion had to be discussed are missing, while other parts poorly presented.
I refer mainly to the Section 4 concerning antimicrobial nanomaterials and Section 5 concerning the synergisms.
While in the Introduction and in Figure 3, the authors reported on different type of antimicrobial NPs, in Section 4, as well as in Table 3, they discussed only metallic NPs or similar nanomaterials. The authors should complete this Section discussing all the nanomaterials presented in Figure 3, including a Table like Table 3 for each class of antimicrobial NPs discussed. In this regard, relevant reviews published in 2020 and in 2021 on Polymers (MDPI) and Nanomaterials (MDPI) could be of help for discussing antimicrobial polymers and dendrimers.
Section 5 is very poor, and much more case studies on the synergistic action of combined antimicrobial devices should be reported. In this regard, I think essential to report on the combinations, both already clinically approved and in experimentation, of antibiotics and β-lactamase inhibitors. In this regard, I suggest authors to read two relevant reviews published in 2022 on Pharmaceuticals (MDPI).
I suggest authors to insert also the information reported in Section 5 in a Table.
In my opinion, a Section in which the current conventional available antibiotics clinically applied are discussed is mandatory. Also in this new Section, a Table summarizing the several families of antibiotics, with their resistance profiles and the target bacteria would be desired.
Another but not the last concern about this paper is the use of the English language. There are many grammatical errors, verbs are often used incorrectly, and some typos must be solved. The paper needs a revision by an English expert who should supply a certificate attesting his/her revision.
Following only some examples.
Line 49. “an” must be removed.
Line 106. The comma in 28,000 is missing.
Line 112. “Metabolites are” and not “is”.
Line 114. “produce” not “produces”.
Line 117. “are” is necessary before “classified”.
Line 129. “consisting of” in place of “consist of”.
Line 132. “have” not “has”
Line 137. “acting” in place of “act”.
Line 139. “are” is a typos and must be removed.
Additionally, some sentences are not clear and need reformulation.
The captions of Figure 1 and Figure 2 are incorrect and need to be reformulated like that of Figure 3.
Are the authors sure that essential oils are colourless??? (Line 181).
Are the authors sure that phytochemical haven’t side effects? I am more confident that side effects depend strongly on concentrations, and often antibacterial concentrations are very high…
On these considerations, this study supplies relevant but incomplete information, which in addition need of a better organization and presentation. I ask authors to address all the above-mentioned issues for making the present paper worthy of further consideration for publication on Biomedicines.
Author Response
Dear Editor and Reviewers
Manuscript ID: biomedicines-1870450
Dear Editor and reviewers
Thank you very much for your kind consideration and invaluable efforts to review our manuscript. We are truly grateful for the meaningful comments and insights of the reviewers and editor. We have carefully revised all issues raised by the reviewers. We feel that our manuscript has been significantly improved from the reviewer’s creative and insightful suggestions. We would like to submit our fully revised manuscript after revised of all comments. All corrections made are highlighted in green color in the revised manuscript. Please see the response to reviewers’ comments on the succeeding pages.
Thank you very much for your consideration.
Sincerely Yours,
Kwang-Hyun Baek
Reviewer 2
Considering the importance of finding new and effective therapeutic strategies to counteract the currently untreatable infections sustained by MDR pathogens and by biofilms producing bacteria and the so-called biomaterials-associated nosocomial infections, the topic of the present review is original and interesting. Anyway, the way the authors have afforded the subject, is poor and badly organized. Essential parts, which in my opinion had to be discussed are missing, while other parts poorly presented.
Comment 1
I refer mainly to the Section 4 concerning antimicrobial nanomaterials and Section 5 concerning the synergisms.
While in the Introduction and in Figure 3, the authors reported on different type of antimicrobial NPs, in Section 4, as well as in Table 3, they discussed only metallic NPs or similar nanomaterials. The authors should complete this Section discussing all the nanomaterials presented in Figure 3, including a Table like Table 3 for each class of antimicrobial NPs discussed. In this regard, relevant reviews published in 2020 and in 2021 on Polymers (MDPI) and Nanomaterials (MDPI) could be of help for discussing antimicrobial polymers and dendrimers.
Response
We are thankful for the reviewer note. As suggested by reviewer, we have included and discussed all the nanomaterials presented in Figure 3 and included Table 4, 5, 6 and we discussed according to relevant review papers of 2020 polymers and 2021 Nanomaterials. Kindly see below Tables 4, 5, and 6 in the revised manuscript
Core-shell quantum dots (CSQDs) are a new type of fluorescent antibacterial nanomaterial with unique physical and chemical properties. Owing to their high electron transfer, CSQDs produce a large number of free electrons and holes that accumulate ROS inside the cell, inhibiting their respiration and replication [168]. CSQDs exert antimicrobial effects by destroying cell walls, binding with genetic material, and inhibiting energy production. The antimicrobial activities of some CSQDs and their mechanisms of action are listed in Table 4.
Table 4. Antimicrobial activities of core-shell quantum dots against various pathogens with their respective mechanisms of action.
NPs |
Size (nm) |
Bacteria |
Modes of Action |
Ref |
ZnS and CdSe/ZnS quantum dot |
1.9 |
E.coli, B. subtilis |
toxic composition of CdSe QDs they demonstrate antimicrobial behavior |
[169] |
CdSe/CdS/ZnS Multi-Core–Shell quantum dots |
12-38 |
K. Pneumoniae, P. aeruginosa |
rupturing of the membrane wall and cause of the decay of bacteria |
[170] |
Ag-PdS/ZnS/CdS core/shell quantum dots |
8 |
S. saprophyticus, E. coli |
establishment of the catalyst–microorganism complex and a catalyst-related ROS |
[171] |
ZnSe@ZnS core-shell quantum dots |
3.6 & 4.8 |
E. coli, S. aureus |
high affinity towards the thiol groups of bacterial cell surface proteins |
[172] |
Peptide-loaded CdSe quantum dot |
9 & 14 |
E. coli, S. aureus |
AP loaded on CdSe NPs had a higher water solubility and bioavailability
|
[173] |
P-doped carbon quantum dots |
2.75-4.25 |
E. coli, S. aureus |
cell walls wrinkled and broken |
[174] |
Ag@Ag2O core–shell |
19-60 |
P. aeruginosa, S. aureus |
blockage of DNA replication and repair processes |
[175] |
Additionally, the antibacterial properties of graphene involve both chemical and physical modes of action. The chemical action is associated with oxidative stress generated by charge transfer and ROS, while the physical action is induced by the direct contact of graphene with bacterial membranes [176]. Similarly, CNTs are more effective, cost-efficient, and exhibit strong antimicrobial properties owing to their remarkable structure. This mechanism is based on the interaction of CNTs with microorganisms and the disruption of their metabolic processes, cellular membranes, and morphology. Table 5 summarizes the various reported antimicrobial activities of graphene and CNTs.
Table 5. Antimicrobial activities of graphene and CNTs against various pathogens and their corresponding mechanisms of action.
NPs |
Size (nm) |
Bacteria |
Modes of action |
Ref |
rGO-TiO2 |
32 |
E. coli, S. aureus |
improve the contact between TiO2 surface and bacteria |
[177] |
GO-ZnO |
14–26 |
E. coli |
s induce ROS to kill the bacteria |
[178] |
GO-Cu2O |
30 |
E. coli, S. aureus |
copper ions react with cyto- plasmic constituents |
[179] |
DMS-GO-DMA |
-- |
E. coli, S. aureus |
GO induce membrane stress on-contact by disrupting and damaging cell membranes |
[180] |
MWCNT-LVX |
-- |
S. aureus, P. aeruginosa |
inhibition of bacterial DNA replication |
[181] |
F-MWNTs |
-- |
E. coli, S. aureus |
smaller diameter of MWNTs can endorse damage to cell membrane through the cell-surface interaction |
[182] |
Ag-doped ZnO on SWCNTs |
12-15 |
E. coli, S. aureus |
production of ROS on the interaction samples with bacterial membrane |
[183] |
Au-doped ZnO on MWCNTs |
12-18 |
E. coli, S. aureus |
The toxicity of carbon nanotube is mainly affected by diameter, length, surface functional group |
[183] |
Dendrimers are macromolecules with highly branched tree-like dendritic structures, narrow sizes, relatively large molecular masses, and well-defined globular structures [184]. Dendrimers are peripherally cationic and highly water-soluble, due to numerous peripheral hydrophilic groups compatible with water [185]. Dendrimers can incorporate biologically active agents in the interior or periphery; therefore, they serve as carriers of biologically active agents [186]. Antimicrobial polymers or their composites can prevent or suppress the growth of microbes on their surfaces or in the environment. Positively charged polymer surface groups are attracted to negatively charged cell membranes, leading to cell membrane damage and cell death [187]. A few recent publications on the antimicrobial activities of dendrimers and polymer nanocomposites are summarized in Table 6.
Table 6. Antimicrobial activities of dendrimers and polymer composites against various pathogens with their mechanisms of action.
NPs |
Size (nm) |
Bacteria |
Modes of action |
Ref |
Van-PAMAM-AgNP dendrimers |
-- |
S. aureus |
heterofunctionalized Van-PAMAM-AgNP dendrimers for intra-cellular entry through the cell wall and bacterial killing |
[186] |
G4-PAMAM dendrimer |
10 |
E.coli, B. subtilis |
disrupting of the cell membrane function and inhibiting cell wall synthesis, nucleic acid synthesis, protein synthesis |
[188] |
PAMAM-G7 dendrimer |
20 |
P. mirabilis, S. aureus |
dendrimers are mediated by disrupting the bacterial outer and inner membrane by terminal amine groups |
[189] |
Amino acid-modified polycationic dendrimers |
-- |
P. aeruginosa |
loss of membrane potential, inhibition of biosynthetic pathways, and free radical production |
[190] |
Triclosan-loaded polymeric composite |
-- |
S. aureus, K. pneumoniae |
at high concentrations, triclosan destroys the bacterial membrane, leading to its death |
[191] |
PBAT/Cu-NPs |
100-200 |
A. baumannii, E. faecalis |
polymer and metal nanocomposites increase the number of ions released from the nanoparticles into the polymer matrix |
[192] |
Piperazine polymer nanocomposite |
559.7 |
E. coli, S. aureus |
nanoparticles are distributed within the suitable polymer matrix |
[193] |
PVA/GO/Ag nanocomposites |
-- |
E. coli, S. aureus |
physical interactions of the bacterial cell with the nanoparticle |
[194] |
Comment 2
Section 5 is very poor, and much more case studies on the synergistic action of combined antimicrobial devices should be reported. In this regard, I think essential to report on the combinations, both already clinically approved and in experimentation, of antibiotics and β-lactamase inhibitors. In this regard, I suggest authors to read two relevant reviews published in 2022 on Pharmaceuticals (MDPI).
I suggest authors to insert also the information reported in Section 5 in a Table.
Response
We are grateful to the reviewer. As suggested by the reviewer, we discussed more case studies on the synergistic action of combined antimicrobial actions including antibiotics and β-lactamase inhibitors by referring to two relevant reviews of Pharmaceutics 2022. We have included one more Table for section 5. Kindly see below or Line 360-395 and Table 7 in the revised manuscript.
Some combinations of antimicrobial agents, such as plant extracts, EOs, antibiotics, and NPs, are summarized in Table 7.
Table 7. Antimicrobial activity of combinations of plant extract, EOs, antibiotics, and NPs against different pathogens.
Antimicrobial agents |
Combinations |
Pathogens |
Ref. |
EOs/EOs |
Melaleuca alternifolia/ Cupressus sempervirens |
E. coli |
[196] |
EOs/Antibiotics |
Eucalyptus globulus/oxacillin |
S. aureus |
[196] |
EOs/NPs |
Lemongrass/Chitosan NP |
E. coli, S. aureus |
[198] |
Plant extract/antibiotics |
Salvadora persica/ amoxicillin |
P. gingivalis, T. forsythia |
[199] |
Plant extract/EOs |
Origanum vulgare/ carvacrol |
S. aureus |
[200] |
Plant extract/NPs |
Vatica diospyroides/Ag NPs |
S. aureus, B. subtilis |
[201] |
NPs/antibiotics |
AgNPs/fluconazole |
S. aureus, E. coli, |
[202] |
β-Lactam/β-Lactamase inhibitor |
amoxicillin / potassium clavulanate |
S. aureus |
[203] |
Combinations of antimicrobial agents, such as EOs/EOs, plant extract/plant extract, and NPs/NPs, already have confirmed antimicrobial activities [204]. Ncube et al. evaluated the bulb and leaf extracts of three medicinal plants, independently and in combination, against S. aureus. Their results showed the strongest synergistic effect compared with the effects observed with individual extracts [12]. Similarly, a combination of Bulbine frutescens and Vernonia lasiopus plant extracts showed improved antimicrobial activity against E. coli [205]. Obuekwe et al. found the largest zones of inhibition against S. aureus using a combination of Ocimum gratissimum and Ficus exasperate, and Bryophyllum pinnatum and Ocimum gratissimum against E. coli [206]. Recently, EO–EO associations showed a synergistic effect against vancomycin-resistant enterococci (VRE), methicillin-resistant S. aureus (MRSA), and extended-spectrum β-lactamase (ESBL)-producing Escherichia coli [196]. A mixture of R. abyssinicus and D. penninervium EOs showed strong synergistic effects against MRSA and P. aeruginosa [207].
The synergistic antibacterial activities of cumin, cardamom, and dill weed EOs against C. coli and C. jejuni have been reported [91]. In a previous study, a combination of cinnamon and clove EOs showed synergistic antibacterial activity against foodborne S. aureus, L. monocytogenes, S. typhimurium, and P. aeruginosa [208]. Garza-Cervantes et al. examined the synergistic antimicrobial activities of silver in combination with other transition metals (Zn, Co, Cd, Ni, and Cu). Their results exhibited synergism since the antimicrobial effects of the combinations against E. coli and B. subtilis increased up to 8-fold when compared to the individual metals [209]. Similarly, β-lactam is the most common bactericidal agent recommended for the treatment of several infectious diseases. However, the increasing emergence of β-lactam resistance due to β-lactamase enzyme production is one of the most serious public health threats. Hence, current clinical trials suggest the use of proper combinations of β-lactam and β-lactamase inhibitors [210]. β-lactam inhibitors are associated with β-lactam antibiotics because they are hydrolyzed by β-lactamases, and their main objective is to protect the associated antibiotics. β-lactam inhibitors prevent the hydrolytic action of β-lactam antibiotics by binding to the active site of β-lactamase enzymes [211].
Comment 3
In my opinion, a Section in which the current conventional available antibiotics clinically applied are discussed is mandatory. Also in this new Section, a Table summarizing the several families of antibiotics, with their resistance profiles and the target bacteria would be desired.
Response
We are highly grateful to the reviewer for highlighting this important point. As per the reviewer's suggestion, we have included one more section to discuss currently available conventional antibiotics and also included Table 8 to summarize several families of antibiotics with their resistance profiles and the target bacteria. Kindly see the below or Line number 397-415 or section 6 in the revised manuscript.
- Currently available conventional antibiotics
Currently, there are 32 antibacterial agents in clinical development phases 1-3 targeting WHO priority pathogens, 12 of which have activity against gram-negative pathogens. Since 2018, several new products have entered phase 1 trials, and two new recombinant topoisomerase inhibitors, zoliflodacin and gepotidacin, have moved from phase 2 to phase 3 trials. Similarly, lefamulin and relebactum moved from phase 3 trials to FDA approval, and omadacycline and eravacycline have moved from NDA submission to gaining FDA approval. An additional product of β-lactam (cefideocol), is more stable against a variety of β-lactamases and has activity against all three critical priority pathogens [212]. β-lactams are well-established and widely used antibiotics, including penicillins, cephalosporins, carbapenems, and monobactams. β-lactams interrupt cell wall formation and subsequently disrupt peptidoglycan biosynthesis. However, the emergence of bacteria that produce β-lactamase enzymes that hydrolyze β-lactam antibiotics has rendered many antimicrobial agents ineffective. β-lactamases are a diverse class of enzymes produced by bacteria that break the β-lactam ring open, inactivating the antibiotic. Table 8 summarizes several mechanisms of resistance to different target drugs with different modes of action.
Table 8. Mode of action of different classes of antibiotics with their resistance profiles and target bacteria.
Antibiotics |
Specific drug |
Modes of action |
Resistance profiles |
Target bacteria |
Ref. |
β-Lactams |
Penicillin G, amoxicillin, and cephalosporin C |
Cell wall synthesis inhibition |
Hydrolysis, efflux, altered target, reduced permeability |
S. aureus, P. aeruginosa |
[213] |
Aminoglycosides |
Streptomycin, gentamicin |
Inhibition of translation and cell membrane synthesis |
Modifying enzyme inactivation by phosphorylation |
P. aeruginosa, V. cholerae |
[214] |
Tetracyclines |
Minocycline, Doxycycline |
30S ribosomal subunit |
Monooxygenation, ribosomal modification |
Staphylococci, Streptococci |
[215] |
Glycopeptides |
Vancomycin, Teicoplanin |
Peptidoglycan biosynthesis |
Altered target |
S. haemolyticus, E. faecium |
[215] |
Macrolides |
Erythromycin, azithromycin |
Inhibition of protein synthesis |
Glycosylation, efflux, methylation |
Streptococci, Staphylococci |
[216] |
Phenicols |
Chloramphenicol |
Inhibition of protein synthesis |
Acetylation by chloramphenicol acetyltransferase |
B. subtilis, S. pneumoniae |
[217] |
Rifamycin |
Rifampin |
Inhibition of nucleic acid synthesis |
ADP-ribosylation, efflux |
V. cholerae, E. coli, |
[214] |
Quinolone |
Ciprofloxacin, levofloxacin |
Inhibitors of DNA synthesis |
Altered DNA gyrase |
S. aureus, P. aeruginosa |
[213] |
Cationic peptides |
Polymyxin B, colistin |
Disrupt membranes |
Altered target, efflux |
E. coli, S. typhimurium |
[214] |
Comment 4
Another but not the last concern about this paper is the use of the English language. There are many grammatical errors, verbs are often used incorrectly, and some typos must be solved. The paper needs a revision by an English expert who should supply a certificate attesting his/her revision.
Following only some examples.
Line 49. “an” must be removed.
Line 106. The comma in 28,000 is missing.
Line 112. “Metabolites are” and not “is”.
Line 114. “produce” not “produces”.
Line 117. “are” is necessary before “classified”.
Line 129. “consisting of” in place of “consist of”.
Line 132. “have” not “has”
Line 137. “acting” in place of “act”.
Line 139. “are” is a typos and must be removed.
Response
We are thankful for the reviewer's note. As suggested by the reviewer we have revised our whole manuscript by Editage, a division of Cactus Communications. We have attached a certificate of English editing.
Comment 5
Additionally, some sentences are not clear and need reformulation.
The captions of Figure 1 and Figure 2 are incorrect and need to be reformulated like that of Figure 3.
Response
We are highly grateful to the reviewer for highlighting this important point. The captions of Figure 1 and Figure 2 is corrected. Kindly see below or in the revised manuscript
Figure 1. The extracts of plant organs namely the root, bark, bulbs, leaf, flower, and seed, may encompass distinctive phytochemicals with antimicrobial properties
Figure 2. Schematic representation of some important EOs contain various components which have been screened for their antimicrobial properties
Comment 6
Are the authors sure that essential oils are colourless??? (Line 181).
Response
As suggested by the reviewer we have corrected the revised manuscript.
Comment 7
Are the authors sure that phytochemical haven’t side effects? I am more confident that side effects depend strongly on concentrations, and often antibacterial concentrations are very high
Response
As suggested by the reviewer we have corrected the revised manuscript. Kindly see below or in the revised manuscript
fewer side effects
mitigate side effects to the body with lower drug concentrations

Round 2
Reviewer 2 Report
Dear Authors,
You have done a great work of Revision and I am now sattisfy with you Review that I consider suitable for publication in this form.
Congratulation